# Conflict between Science and Superstition in Medical Practices

Donat Uwayezu [1], Eustache Ntigura [1], Agnes Gatarayiha [1], Anna Sarah Erem [2], Mainul Haque [3], Md Anwarul Azim Majumder [4] and Mohammed S. Razzaque [1,5,*]

1 Department of Preventive & Community Dentistry, School of Dentistry, University of Rwanda, Kigali P.O. Box 3286, Rwanda
2 Department of Pathology and Laboratory Medicine, School of Medicine, Emory University, Atlanta, GA 30322, USA
3 Unit of Pharmacology, Faculty of Medicine and Defense, National Defense University of Malaysia, Kuala Lumpur 57000, Malaysia
4 Faculty of Medical Sciences, Cave Hill Campus, The University of the West Indies, P.O. Box 64, Bridgetown BB11000, Barbados
5 Department of Pathology, Lake Erie College of Osteopathic Medicine, Erie, PA 16509, USA
* Correspondence: mrazzaque@lecom.edu

**Abstract:** Superstition is a belief that is not based on scientific knowledge. Traditional healers usually use superstition in their practices to manage human health problems and diseases; such practices create a conflict with the medical profession and its evidence-based practices. Medical professionals confirm that this kind of practice is unsafe as it is performed by untrained people (e.g., traditional healers) utilizing unsterilized instruments within unhygienic environments. Most of the cases eventually develop a variety of complications, which are sometimes fatal. Female genital mutilation, uvulectomy, oral mutilation (tooth bud extraction to cure "Ibyinyo"), and eyebrow incisions are examples of the many different types of superstitious practices which occur commonly in other parts of the world. We describe how these traditional practices of superstition have been and continue to be performed in various parts of the world, their complications on oral and general health, and how such practices hinder modern medical practices and highlight huge inequalities and disparities in healthcare-seeking behavior among different social groups. This paper aims to increase health literacy and awareness of these superstition-driven traditional and potentially harmful practices by promoting the importance of evidence-based medical practices.

**Keywords:** superstition; traditional healers; clinical practice; conflict; awareness; socioeconomic status

## 1. Introduction

Science can be broadly defined as an organized system for collecting knowledge with the ultimate goal of understanding the nature of the world and the universe. It is based on the assumption that there is an objective reality to the universe that a rational mind can observe. These observable truths are the facts that form the basis for rational assumptions [1,2]. Historically, science consisted simply of making observations to gain a deeper understanding of natural phenomena. In modern science, the laws of nature are determined through a rigorous process of scientific analysis. Experiments are meticulously conducted to test a hypothesis based on existing knowledge, and data is recorded in detail to evaluate it for accuracy and reproducibility by others [1,3].

In the modern day, medicine is considered by many to be a branch of science [4]. The term "Medical science" is broad and encompasses a myriad of disciplines, covering both basic science (knowledge of the normal functions of the human body and the etiology of diseases) and applied science (methods and instruments for diagnosing, treating, and preventing diseases). Medical research is the process of conducting experiments to expand knowledge of human health and diseases to develop and improve upon existing practices.

The goal is to use empirical evidence to guide standards that maximize beneficial health outcomes and minimize potential harm from medical practices [2,5].

Historically, medicine was practiced more as an art with connections to religious or philosophical practices. Today, however, medicine follows the same rigorous principles of observation, experimentation, and evaluation as other areas of science. Evidence-based medicine is now the gold standard for medical and dental practices, and many healthcare monitoring organizations, including the United States Food and Drug Administration (FDA) and the European Commission, require strong scientific evidence for the safety and efficacy of drugs, medical devices, and procedures before approving them for use in their respective countries. Similarly, healthcare practitioners licensed in most countries must receive accredited evidence-based medical education [2,4,6].

## 2. Superstition

Superstition is the non-scientific belief in magical connection. Such beliefs include aspects of certain religions and supernatural ideologies such as astrology, omens, witchcraft, and prophecies [7]. These are notions still maintained by specific populations despite evidence to the contrary. An example of a superstition is the concoction of a mystical recipe to ward off evil spirits that cause disease [8]. Even after all the modernization and enlightenment that has taken place, superstitious beliefs persist in our societies. It is thought that most human beings, to a certain level, have irrational, superstitious views [9,10]. According to the World Health Organization, traditional healing has been widely practiced in developing countries, mainly by people of lower socioeconomic status. Approximately 80% of the patients depend on traditional healing practices to treat common diseases. A study conducted in rural Bangladesh found that traditional healing methods are widely practiced, as evidence-based health care is either expensive or sometimes non-existent [11].

Historically, superstition has chiefly stemmed from illiteracy, immaturity, and anxiety about the unknown, usually related to the incorrect clarification of natural events (Figure 1). Superstitions could have a religious, cultural, and personal basis [8]. Of relevance, religious principles and practices may appear superstitious to a person without faith. Many communities have had an enormous and diverse array of cultural and ritual superstitions throughout history. Many cultures have held illogical views concerning ways to ward off illness, bring good fortune, predict the future, avert disease or accidents, and even choose a mate [12]. Even a medical student who does well in a professional examination may begin to think their success is due to the use of a specific pen and that the pen, therefore, is lucky; gamblers win a few times after betting on a black horse and start believing that black horses run well for them [13]. Such irrational justifications are partly based on not trusting that individuals have complete control of events that involve them; a lack of self-belief could lead to fear and faith being put in falsehoods to feel safe and secure [9]. Some superstitious beliefs stem from misunderstandings of natural physiological processes. For example, in many communities, female menstruation was (and in some cases still is) considered "unclean" [14,15]. While menstruating, women were not allowed to visit a temple or touch a pickle (for fear of spoiling it); these menstruating, women also were not allowed to touch other people and commonly had to sit in a separate room and eat from a separate plate from fear of spreading contamination. Even in the UK, many farmers' wives believed that milk handled during menstruation could not be churned into butter or that hams would not take salt from their hands, as it was thought they were profane or unholy [14].

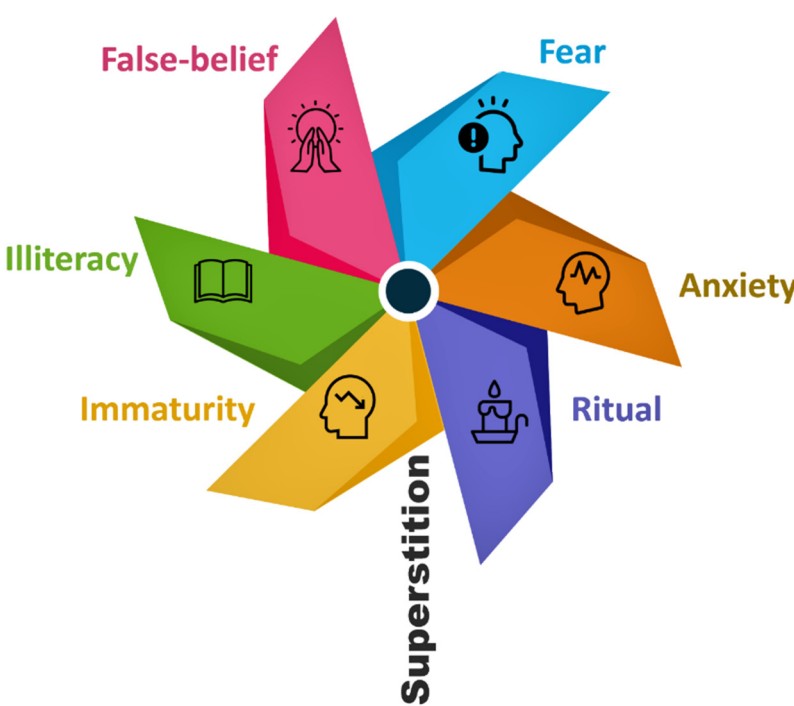

**Figure 1.** A few common causes that lead to superstition-related practices in different parts of the globe that affect oral health to general health.

Some superstitions are derived from an early set of views and thoughts which remain to impact people's activities even though they have lost their basis of evidence through the development of rational thought, based on science. In earlier days, when many occurrences and events were not readily explainable, theories were invented to create seemingly sensible interpretations, such as those involving ghosts and witches. These theories were intended to accommodate the desired reasons for the events which the human mind tries to explain without a scientific basis. Most of these principles and theories became extinct with the advancement of science and technology, and yet, some individuals and societies are still governed or at least prejudiced by these surviving ancient concepts [16].

Scientists, physicians, and researchers have long sought to identify any basis for existing superstitious beliefs and to provide alternative explanations and practices based on rational observation and objective evidence. Solar eclipses offer an excellent example of this. These astronomical events have, for centuries, been the source of anxiety and numerous myths and legends. A solar eclipse is still considered to be a bad omen in several cultures. However, astronomers and behavioral scientists have extensively studied the responses of flora and fauna to solar eclipses. They have determined that eclipses cause no major physical changes to humans, their health, or the environment [7]. Superstitious people are disinclined to wear items of clothing that have been worn by those who have felt pain or otherwise experienced bad luck, such as those who have lost a limb or contracted HIV, or by those with a strong moral flaw, such as a convicted murderer or another type of felon [17,18]. Some also believe that microbial diseases that have caused suffering for their ancestors can also infect them. Contagion thus was explained through shared characteristics or familiarity before the discovery of the microbial basis of disease; even specific dates of the year are presumed to bring bad luck [19,20]. Superstitious views have historically been recognized as the persistent guidance of childhood considerations about the world [17]. These ascend from the rapid, automatic judgments that ordinary people regularly use to make sense of a highly uncertain world [21–24].

### 3. Superstition and Health Care

As mentioned, superstitions can be broadly grouped as religious, cultural, and personal (Figure 2). Though superstition has been declining with the advancement of science, superstitious beliefs still influence medical care. For example, Kyoto University Hospital patients in Japan were shown to want to stay in the hospital beyond what was recommended by doctors. Researchers attributed this behavior to the patient's superstitious beliefs about being released on a lucky day versus an unlucky day. Allowing these superstitious beliefs to influence the decision when to discharge patients from the hospital significantly increased medical care costs in Japan [25,26]. Similarly, a large percentage of the population in less educated Pakistani communities believes in a superstition that controls their health-seeking behavior and factors into how the community responds to health intervention programs. Understanding these superstitious beliefs is imperative to develop effective health programs and ensuring quality healthcare delivery [27,28].

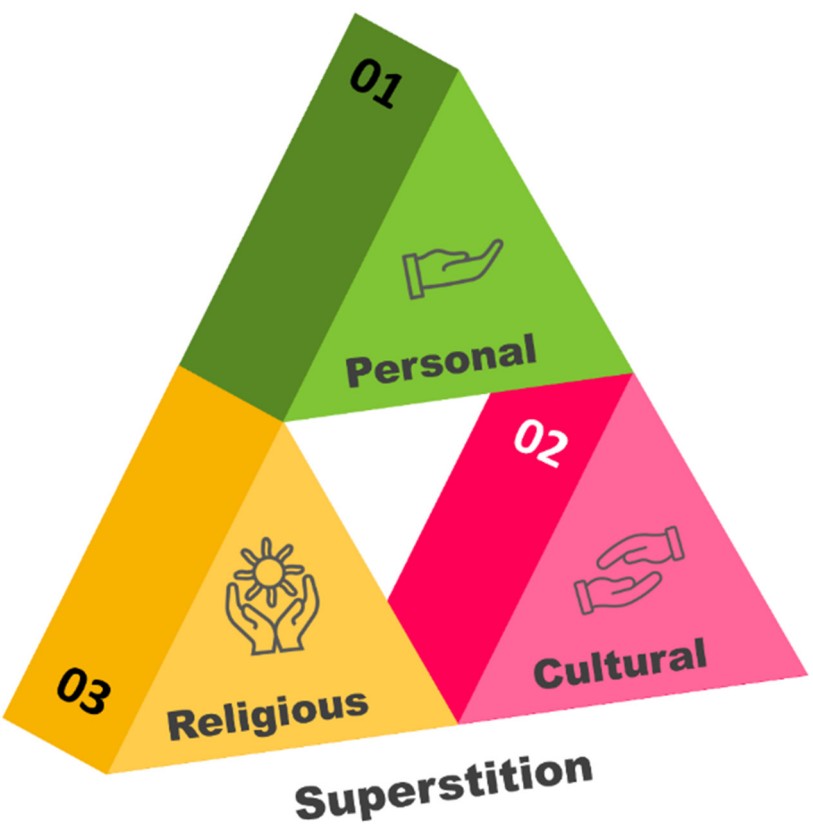

**Figure 2.** Superstition can be broadly based on three different irrational behaviors or beliefs to justify certain unexplained events.

Virtually every human fears illness regardless of gender, religious belief, ethnicity, or education. This fear has led to the development of supernatural practices for treating and preventing disease, where evidence-based medical care was overlooked. In the US state of Louisiana, for example, superstitious beliefs and practices were common in a subset of the population, especially in the southern part of the state, where the voodoo religion was one way of handling illness and fear [29]. Webb (1971) recommended health educational programs and increased public health activities to encourage people to adopt modern healthcare practices and provide protection from these superstitions and voodoo practices. It was also suggested that healthcare professionals needed to develop skills to recognize local cultures and beliefs to deal with unsafe practices to patients' health [29].

Perhaps more so than superstition, religious practices still profoundly influence medical practices and decision-making. For instance, the family members of patients in intensive

care units of three tertiary care hospitals in Greece often practiced a series of spiritual rituals, namely the use of blessed oil and holy water, the use of relics of saints and holy icons, the offering of names for pleas, and pilgrimage [30]. In Pakistan, medical students believe that mental illnesses have paranormal, magical, or mystic root causes and that mental illness can only be treated by religious, ecclesiastical, or faith-based persons [31]. Interestingly, one study found that even in the United States, work-related superstitions among healthcare professionals were prominent, have existed for a long time, and will probably persist into the future [32]. In a 2004 study investigating work-related superstitions, 300 question-naires were mailed to perioperative nurses in the Pittsburgh area. While only 23% of the responders thought of themselves as superstitious, there was widespread belief in certain workplace superstitions. Seventy-eight percent of the responders believed red-headed patients were more likely to have complications such as bleeding [32]. Sixty-nine percent believed that the full moon was associated with a heavier workload, and sixty-six percent believed that certain nurses work under a "black cloud" [32].

Superstitions have long been used to explain illnesses and medical phenomena when scientific rationales were unavailable. Many of these beliefs persist today, particularly among communities and demographics that are not regularly exposed to scientific dis-coveries or education. While most people in communities in India and Pakistan have some type of superstitious beliefs, these are much more common and substantial among illiterate people. Often, superstitions are perpetuated in part by scammers who wish to take financial advantage of people in medical need and who are susceptible to belief in occult powers. These beliefs can negatively impact health care, primarily because people will often visit a traditional or spiritual healer rather than a certified medical professional for treatment [27,33]. Likewise, Ghana's executive director of health has expressed his concern that Ghana's health care system was being held back by the fact that many Ghanaians still believed more in superstition rather than the existence of "germs" and other scientific causes for ailments [34,35].

Understanding these superstitious beliefs has been vital for better health care and medical management in these countries. By addressing the superstitious concerns of people and governments, healthcare providers can conduct better outreach and offer much-needed medical care. Interventions involving community participation have proven effective in overcoming superstitious beliefs, but only when cultural and spiritual beliefs are approached with awareness and sensitivity [33]. A study on mostly low-income cohorts found that superstitious beliefs about epilepsy existed among the participants [36]. It is important to mention that many of these superstitions persist because of inequity in resources between the communities. Access to evidence-based medicine in the form of care or information is related to economic privilege. Economic inequality must be addressed simultaneously to reduce superstitious practices.

## 4. Conflict between Superstition and Clinical Practices

There is a natural conflict between traditional healers whose practices are based on superstition and medical professionals who strictly adhere to evidence-based medical practices. Traditional healers often use unsterilized instruments in unsafe environments that can lead to harmful, potentially fatal impacts on their patients. These practices are rejected by conventional medicine as not being scientifically based [37–40]. Female genital mutilation, uvulectomy, oral mutilation, and eyebrow incisions are examples of harmful superstitious practices that occur worldwide. These unscientific and unsafe practices have had immediate and long-term health consequences depending on the severity and type of procedure performed [41–44]. Patients' most common and immediate outcomes include pain, shock, septicemia, tetanus, hemorrhage, ulceration of the involved and adjacent parts, and even death. Long-term complications include abscesses, keloid scars, cysts, sexual dysfunction, and malformed or missing body parts [45–47].

On rare occasions, an infant might be born with erupted teeth, and in ancient Rome, such infants were thought to be destined for greatness and were dearly referred to as "Dent

at us". On the other hand, certain tribes in Africa, the Middle East, and India considered infants with erupted teeth evil or impure. In a worst-case scenario, some of those infants were killed in the name of protecting the community. Superstition has been identified in many regions worldwide, and people involved in these practices experience many adverse effects on their oral and general health [48–50]. A study conducted in Ethiopia on the impact of female genital mutilation showed that respondents to a questionnaire experienced complications at the following rates: excessive bleeding during the procedure (55%), urine retention (36%), infection (11%), and swelling of the genitalia (11%) [51]. Contrarily, the medical benefits of male circumcision have been widely studied and reported [52]. Clinical trials have found that male circumcision reduced the risk of acquiring genital herpes by around 30%, the risk of developing genital ulceration by 47%, and HIV acquisition in men by 51% to 60% [53]. Although the benefits of male circumcision appear overwhelming, and in modern hospital settings, the surgical risk of male circumcision is negligible, in certain countries, traditional healers perform male circumcision in rural areas, which may be a cause of health concerns. Another Tanzanian study reported that the prevalence of uvulectomy was around 4% in that region, and the most common indication for uvulectomy was cough (81%). The highest reported complications were severe hemorrhage (66%), rejecting food (9%), and failure to gain weight (9%) [54]. Multiple other studies reporting on the effect of early marriage have shown that girls in the ten-to-fourteen-year age range are five to seven times more likely to die from childbirth than adult women. In contrast, girls between fifteen and nineteen years of age are twice as likely to die compared to women who become married above the age of twenty years old. Birth injuries, serious childhood illness, with mental and physical disabilities are also associated with early marriage-associated childbirth [55–58].

Gukura Ibyinyo or Ebinyo is a form of infant oral mutilation performed in Rwanda and parts of the neighboring East African countries. It is an illegal and unsafe traditional dental procedure that traditional healers and parents have practiced on their young children in most East African countries. Traditional instruments are used to extract deciduous canine tooth buds at around six months of age. Teething babies can experience oral pain, fevers, vomiting, or diarrhea, but in traditional medicine, this is sometimes attributed to a disease called "Ibyinyo." Premature extraction of these teeth by traditional healers is believed to cure this condition. However, this practice can cause severe and permanent damage to the child resulting in missing teeth and the destruction of some of the surrounding permanent teeth [59–63]. A 1989 study conducted in Southern Sudan on the prevalence of Ibyinyo among hospitalized infants found that one deciduous tooth had been removed from all the studied infants ($n = 90$), and most were suffering from dehydration caused by various other abdominal complications [64]. In a rural Kenyan Maasai community, another study conducted on a group of 95 children aged between six months and two years reported that 87% of the children had one or more deciduous canine tooth buds removed. In an older age group (three to seven years of age), 72% of the 111 children examined displayed missing mandibular or maxillary deciduous canines [65]. The results of this study are similar to another one conducted in Uganda, where the mean number of affected teeth per child was 3.8; of relevance, 99% of the extracted teeth were canines [66]. Between 1992 and 1998 in Uganda, the complications of Ibyinyo, including sepsis and anemia, were the leading causes of hospital admission and hospital death (with a case fatality rate of 21% and a proportional mortality rate of 3%) [46].

## 5. Conclusions

Certain traditional practices are a public health concern that contradicts conventional medical professional practices. These can affect the oral and general health of the affected population, depending on the type of practice performed. Such practices are spread to other parts of the world by immigration or settlement. Changing the mindset of the people who keep practicing these old habits is not easy. However, education programs that consider the sensitivity of traditional practices' cultural and religious aspects are the best hope of

eradicating these harmful practices (Figure 3). The following recommendations would likely reduce unprofessional superstition-based harmful practices in society [2,3,67–71]:

- Health literacy and awareness of professional health education and promotion in the communities to prevent these unprofessional traditional practices.
- Medical and other Health professionals should educate the community on the harmful effects of traditional practices through community health outreaches.
- The danger of blood-borne infections such as HIV that may occur due to these practices should be emphasized.
- Availability, accessibility, and quality of professional healthcare services in the community should be improved.
- Harmful consequences of unsafe traditional health practices should be included in all health science curricula.
- Traditional healers should be held accountable for unprofessional acts and practices.
- Develop policies, standards, and regulatory frameworks to safely and effectively use helpful traditional medicine.

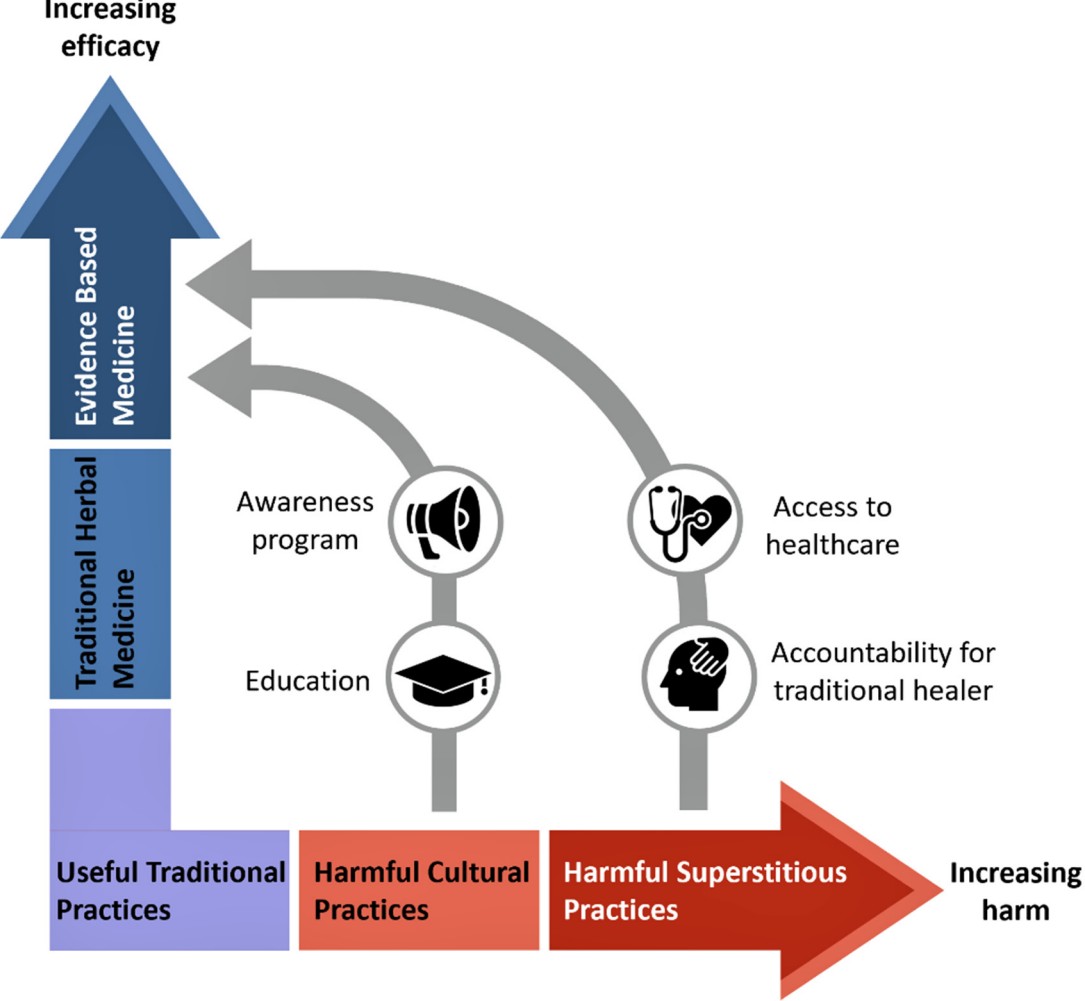

**Figure 3.** Simplified diagram showing the role of education, awareness, and availability of health care in the battle against harmful cultural and superstitious practices. "Kampo" is an excellent example of the useful practice of traditional medicine in Japan [72], while oral mutilation ("Ibyinyo") is an example of harmful cultural practice [73]. The physical and emotional torture of mentally disabled individuals is an example of the harmful superstitious practice.

**Author Contributions:** Conceptualization, M.S.R.; writing—original draft preparation, D.U., E.N. and A.G.; writing—review and editing, A.S.E., M.H., M.A.A.M. and M.S.R.; supervision, M.S.R. All authors have read and agreed to the published version of the manuscript.

**Funding:** This research received no external funding.

**Institutional Review Board Statement:** This is a review article and no research was conducted.

**Informed Consent Statement:** Not applicable.

**Data Availability Statement:** Not applicable.

**Acknowledgments:** The authors want to express their sincere gratitude to Nuraly Akimbekov (Al-Farabi Kazakh National University, Kazakhstan), for his help in drawing the illustrations.

**Conflicts of Interest:** The authors declare no conflict of interest.

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
