# Peer review of "Conflict between Science and Superstition in Medical Practices"

_ime, doi:10.3390/ime1020007_

Round 1
Reviewer 1 Report
This paper tackles an important and neglected problem in medicine, and it is important to bring the problem to the attention of all physicians and medical students. I have just some minor remarks.
1) I wonder why, unlike female genital mutilation, male circumcision is not mentioned in the paper. After all, it is also based on superstition, although it may be less harmful than its female counterpart.
2) Line 48: I don´t quite understand what "the mystic connection" means. Perhaps "belief in magical connections" would be better.
3) Line 54: better: ...most human beings to a certain level believe in...
4) Line 56: better: Superstition results mostly from illiteracy...
5) p. 3, Figure 1: I think the direction of the arrows should be reversed. For example, the arrows go from "Superstition" to the various causes of it. But if this is a causal diagram, the arrows should go from the cause the effect, and the effect is superstition.
6) Line 235: better: The following recommendations would likely reduce unprofessional...
7) p. 7, Figure 2: I wonder why acupuncture is listed as a useful traditional practice. After all, there is barely any evidence for most of its applications, and its foundations are wrong: there is no such thing as a chi/qi energy and there are no meridians in which chi/qi flows (see Singh & Ernst 2008: Trick or Treatment: The Undeniable Facts about Alternative Medicine). In short, I doubt that acupuncture is a good example here.
Author Response
Reviewer 1
This paper tackles an important and neglected problem in medicine, and it is important to bring the problem to the attention of all physicians and medical students. I have just some minor remarks.
=> Thank you for finding our manuscript meaningful.
1) I wonder why, unlike female genital mutilation, male circumcision is not mentioned in the paper. After all, it is also based on superstition, although it may be less harmful than its female counterpart.
=> Discussed.
2) Line 48: I don´t quite understand what "the mystic connection" means. Perhaps "belief in magical connections" would be better.
=> Modified as recommended.
3) Line 54: better: ...most human beings to a certain level believe in...
=> Modified as recommended.
4) Line 56: better: Superstition results mostly from illiteracy...
=> Modified as recommended.
5) p. 3, Figure 1: I think the direction of the arrows should be reversed. For example, the arrows go from "Superstition" to the various causes of it. But if this is a causal diagram, the arrows should go from the cause the effect, and the effect is superstition.
=> Thank you for the valuable suggestion, we’ve revised the Figure.
6) Line 235: better: The following recommendations would likely reduce unprofessional...
=> Modified.
7) p. 7, Figure 2: I wonder why acupuncture is listed as a useful traditional practice. After all, there is barely any evidence for most of its applications, and its foundations are wrong: there is no such thing as a chi/qi energy and there are no meridians in which chi/qi flows (see Singh & Ernst 2008: Trick or Treatment: The Undeniable Facts about Alternative Medicine). In short, I doubt that acupuncture is a good example here.
=> As pointed out, we’ve removed acupuncture as a good example.

Reviewer 2 Report
I think this an interesting commentary, however, what I think is missing is the fact that many of these superstition persist because of inequity in resources between communities. Access to evidence based medicine manifested as care or information is due to economic privilege and in order to decrease superstition that has to be address simultaneously. I would like to see discussed more in the commentary.
Author Response
I think this an interesting commentary, however, what I think is missing is the fact that many of these superstition persist because of inequity in resources between communities. Access to evidence based medicine manifested as care or information is due to economic privilege and in order to decrease superstition that has to be address simultaneously. I would like to see discussed more in the commentary.
=> We want to thank the reviewer for suggesting such an important point. We’ve briefly addressed this issue in our revised manuscript.

Reviewer 3 Report
This paper addresses an important issue between superstition and clinical practice. The history of superstition is carefully described. The relationship between superstition and health hazards is also noted and the prevention of health hazard is discussed.
In my opinion, minor revision is needed for some points.
#1 Superstitious beliefs still influence medical care (P4 line 112). The author cites reference #24 and #25 in order to support the idea, however, they appear to be outdated.
#2 The author state that "In Pakistan, medical students still believe, even today (P4 line 139)". Reference #28 are cited to explain the state; however, the paper was published in 2014. I think that we can not say the paper represent the present.
#3 Several recommendations were presented to reduce unprofessional superstition-based harmful practices in the conclusion part. Are there any references that would support them?
Author Response
This paper addresses an important issue between superstition and clinical practice. The history of superstition is carefully described. The relationship between superstition and health hazards is also noted and the prevention of health hazard is discussed.
In my opinion, minor revision is needed for some points.
#1 Superstitious beliefs still influence medical care (P4 line 112). The author cites reference #24 and #25 in order to support the idea, however, they appear to be outdated.
=> As recommended, we’ve also cited couple of recent references & discuss those in the revised manuscript.
#2 The author state that "In Pakistan, medical students still believe, even today (P4 line 139)". Reference #28 are cited to explain the state; however, the paper was published in 2014. I think that we can not say the paper represent the present.
=> As pointed out, we’ve modified the text.
#3 Several recommendations were presented to reduce unprofessional superstition-based harmful practices in the conclusion part. Are there any references that would support them?
=> As correctly recommended, we have included a few relevant references.
